# Real-Time Navigation in Google Street View^^®^^ Using a Motor Imagery-Based BCI

**DOI:** 10.3390/s23031704

**Published:** 2023-02-03

**Authors:** Liuyin Yang, Marc M. Van Hulle

**Affiliations:** Laboratory for Neuro- & Psychophysiology, Department of Neurosciences, KU Leuven—University of Leuven, B-3000 Leuven, Belgium

**Keywords:** BCI, virtual navigation, motor imagery, system design

## Abstract

Navigation in virtual worlds is ubiquitous in games and other virtual reality (VR) applications and mainly relies on external controllers. As brain–computer interfaces (BCI)s rely on mental control, bypassing traditional neural pathways, they provide to paralyzed users an alternative way to navigate. However, the majority of BCI-based navigation studies adopt cue-based visual paradigms, and the evoked brain responses are encoded into navigation commands. Although robust and accurate, these paradigms are less intuitive and comfortable for navigation compared to imagining limb movements (motor imagery, MI). However, decoding motor imagery from EEG activity is notoriously challenging. Typically, wet electrodes are used to improve EEG signal quality, including a large number of them to discriminate between movements of different limbs, and a cuedbased paradigm is used instead of a self-paced one to maximize decoding performance. Motor BCI applications primarily focus on typing applications or on navigating a wheelchair—the latter raises safety concerns—thereby calling for sensors scanning the environment for obstacles and potentially hazardous scenarios. With the help of new technologies such as virtual reality (VR), vivid graphics can be rendered, providing the user with a safe and immersive experience; and they could be used for navigation purposes, a topic that has yet to be fully explored in the BCI community. In this study, we propose a novel MI-BCI application based on an 8-dry-electrode EEG setup, with which users can explore and navigate in Google Street View^^®^^. We pay attention to system design to address the lower performance of the MI decoder due to the dry electrodes’ lower signal quality and the small number of electrodes. Specifically, we restricted the number of navigation commands by using a novel middle-level control scheme and avoided decoder mistakes by introducing eye blinks as a control signal in different navigation stages. Both offline and online experiments were conducted with 20 healthy subjects. The results showed acceptable performance, even given the limitations of the EEG set-up, which we attribute to the design of the BCI application. The study suggests the use of MI-BCI in future games and VR applications for consumers and patients temporarily or permanently devoid of muscle control.

## 1. Introduction

A brain–computer interface (BCI) can establish a direct link between the user’s brain activity and an external device, providing an alternative means of communication [1] or interaction with the external world [2]. Electroencephalography (EEG), recorded non-invasively by placing electrodes on the scalp, is often used to record brain activity, from which specific signatures in response to a BCI paradigm are extracted and used to decode the user’s intentions. In order to issue navigation commands, visual paradigms, such as P300 [3] and SSVEP [4], are most commonly used to select navigation targets displayed on a screen, or alternatively, by imagined limb motor activity to control an actuator [5]. An example of the former is a visual BCI to navigate a wheelchair [6,7] or a quadcopter [8] in a real environment. An example of the latter is to use a 5-class motor imagery (MI) BCI for piloting a wheelchair [9]. As for navigation in a virtual environment (VE), some attempts have been made to demonstrate the potential use of a BCI in games [10] and mouse control [11]. In [12], a cave automatic virtual environment (CAVE) was implemented, in which subjects navigate in a virtual street projected on active screens surrounding the subject. However, little effort has been put into developing practical and immersive VE navigation applications. Instead, more attention has been directed towards increasing performance on benchmark datasets by using increasingly advanced decoding algorithms.

We propose a new middle-level control paradigm that offers a good balance between user control and efficiency. It is applicable to both known and unknown scenes. We opted for an application in Google Street View^®^, as users can visit a museum using low-level control, or a new city using middle-level control. To the best of the authors’ knowledge, this is the first BCI navigation application beyond concept demonstrations that provides users with a virtual immersive experience.

The paper starts with a literature review on BCI navigation, including its drawbacks and limitations. The proposed BCI system, including the paradigm, experiment setup, decoders, and user interface, are covered in the Methods and System Design section, which is followed by Results and Discussion sections and the Conclusions.

## 2. Literature Review

The essence of BCI navigation is to extract user’s movement commands from brain activity, primarily from EEG recorded with dry or wet electrodes. Compared to wet electrodes, dry electrodes are less expensive and more convenient, as they do not require gel to secure good galvanic contact with the scalp; however, current solutions feature few electrodes and provide low signal-to-noise ratios (SNRs) [13]. The two commonly used BCI paradigms are based on visually evoked responses or on changes in sensorimotor rhythms (SMRs). The P300 and the steady-state visual evoked paradigms (SSVEP) are the most widely used visual paradigms. P300 is an event-related potential evoked by presenting a rarely occurring stimulus [3], whereas SSVEP is a periodic response evoked by a fixed-frequency visual stimulus [4]. They are used to select one of several displayed targets (i.e., corresponding to navigation commands) [6,7,14,15,16,17,18]; since users are cued by the visual stimuli, they are examples of so-called synchronous BCIs. On the other hand, SMR can be asynchronous, as the changes in mu and beta band power are elicited by self-paced imagined limb movements [19]. Although SMR-based BCI navigation is considered to be more intuitive than synchronous BCI, and does not require the user to attend to visual stimuli, it is less accurate and supports fewer commands [20].

As for the navigation control mode, low- and high-level control are generally considered different approaches suited for different scenarios [20]. Low-level control translates each motion intention to a specific motion command, a series of which forms a continuous movement, providing the user with complete control, even in unknown scenes, albeit this is slow and inefficient. In high-level control, humans have no control over individual movements; the user only selects the destination, and an intelligent agent is responsible for getting there, making control fast and efficient but limited to predefined destinations. Aside from these two control modes, some studies use control methods tailored to specific applications. For example, in [19], the authors proposed a navigation paradigm with two mental tasks to move through fixed paths in a virtual apartment. The right/left-hand motor imagery (MI) enabled subjects to select two different commands at each junction out of three: turn right, turn left, and move forward. According to the latest review paper published in 2022 [20], between 2016 and 2020, about 23 papers were published; the majority were based on SSVEP and P300, with 7 being on SMR. Regarding navigation settings, 19 out of 23 studies considered wheelchair or robot control, and only 4 fully focused on developing immersive virtual navigation applications. In [21], a VR based boating game was developed with one-direction control. In [22], a P300-based BCI was designed to control a virtual wheelchair. A maze game was built in [23] with a single-channel SSVEP-based BCI, whereas in [24] the authors used MI to control forward and backward navigation in a 3D game. In addition, some practical applications have been proposed as well, for example, to have an immersive tour in a virtual library [25], controlled by a 2-task MI BCI. As for the EEG recording techniques, the vast majority relied on wet electrodes, which provide higher signal quality, and only a few used wireless dry electrodes [20]. Another observation is that less attention has been given to VE navigation. EEG-based navigation in a virtual world is still fairly simple. It foregoes immersion, and little effort has been made to create applications that can be transferred to existing ones. In addition, as a more intuitive navigation control paradigm, MI faces some challenging problems, e.g., poor decoder performance and fewer control commands, prohibiting the development of more advanced VE-MI applications. Moreover, low and high-level control are two primarily used control approaches which have limitations in VE navigation, where unknown scenes are more frequent. Thus, low-level control is inefficient, and high-level control is unusable when applied to unknown scenes. Therefore, we propose a system with the following features to target the above problems in this study:Navigation in an immersive environment based on Google Street View allows users to appreciate art in a museum or explore a city.A self-paced BCI based on motor imagery with dry electrodes, making the system more intuitive, user-friendly, and convenient.A new middle-level control approach enables efficient navigation in unknown virtual scenes, associated with error-control strategies that enable the system to be used despite inferior decoding accuracy.

## 3. Methods and System Design

### 3.1. System Overview

The proposed system is illustrated in Figure 1. It consists of an offline training session (left panel) and an online testing session (right panel). In the offline training session, participants follow the displayed instructions and perform IM tasks, which are described in the Participants and BCI Paradigm section. The EEG data are recorded and used for training the decoder, which in turn is used for command classification in the online experiments. The EEG processing pipeline and decoder architecture are discussed in the BCI Decoder section. In the online testing session, the decoder classifies the three IM tasks in real time. The classification output is used by the control logic together with the eye blink detection output to control the navigation in Google Street View^^®^^, which provides direct feedback to the participants through the navigation interface. The interface is explained in the Application Interface section and the control logic in the Navigation Mode section. The online experiment procedures and goals are covered in detail in the Online Experiments section.

### 3.2. EEG Recording

The navigation system we propose is based on imagined movement (IM) eliciting a power increase (event-related synchronization ERS) and decrease (event-related desynchronization ERD) in mu (8–12 Hz) and beta (14–25 Hz) bands in EEG recorded over the ipsi- and contralateral sensorimotor cortex, respectively [26]. The EEG recording device used in this study was Mentalab [27], an 8-channel dry electrode wireless headset. The electrodes are located as marked in red in Figure 2 and re-referenced to the average of both mastoid electrodes (TP9, TP10). The sampling rate was set to 250 Hz.

### 3.3. Implementation Details

The experimental interface and the decoder were programmed in MATLAB App Designer [28]. The recorded EEG data were streamed to the laboratory streaming layer  [29], where it was synchronized to markers. The control logic was implemented on top of the decoder, which sent navigation commands to the Google street view^®^ application through the TCP (Transmission Control Protocol). As the commands involve pressing virtual arrows, the control logic can be adjusted to other navigation applications. The junction detection and navigation application was programmed in Python version 3.7.0 (open-source, downloaded from https://www.python.org/ (accessed on 1 May 2022). To speed up training the decoder, an NVIDIA RTX 2060 GPU was used.

### 3.4. Participants and BCI Paradigm

We recruited 21 healthy adults (6 males, 15 females) without neurological complaints, aged 18 and 31 years. All participants were informed about the procedures and purpose of the experiment and gave their written consent before participating. Experiments were pre-approved by the Ethical Committee UZLeuven, Belgium.

This study used two types of IMs for walking and turning: a 6 s, self-paced, two-hand clenching action to start walking, and a cued-based, 2 s, left or right-hand clenching action to turn left or right (see Navigation Control in Navigation Mode section for detailed control logic). The participants were instructed to sit still; limit eye blinks; and minimize bodily, facial, and arm movements during the movement imagery tasks. The offline experiment was a cue-based training session; the experimental procedure is shown in Figure 3. First, the instruction ("Flexion" or "Left" or "Right") for the upcoming task was displayed for 2 s. Afterwards, an auditory cue (a short 500 Hz tone) was played, indicating the start of the task. Depending on the displayed instruction, the participant imagined a 6 s two-hand clenching action with slow finger flexion or 2 s of left or right-hand clenching, terminated by another auditory cue (a short 300 Hz tone), followed by a 4 s break that allowed the subject to blink and rest between two successive tasks.

There were 4 recording sessions in the offline experiment. In the first two sessions, subjects performed imagined two-hand clenching tasks; in the last two sessions, they performed imagined left or right-hand clenching tasks. Between each session, there was a 5 min break. In total, 70 trials per task per subject were collected, of which 50 were used for offline training and the remaining 20 for offline testing.

After approximately 80 min of the offline experiment, a two-hand clenching decoder (flexion–rest) and a left–right clenching decoder (left–right) were developed per participant. An online experiment was carried out to evaluate the navigation application. In the online experiments, subjects looked at a Google Street View^®^ interface (see Navigation interface in Navigation Mode section). The recording device streamed EEG data to the decoding laptop, where the recorded EEG data were band-pass filtered every 0.5 s and were stored in a buffer. Depending on the control logic and the state, as shown in Navigation control in Navigation Mode section, features were extracted and sent to a classifier, as described in the pipeline in Figure 4. When a navigation command was detected, the system performed the corresponding movement, and based on the control logic, see Navigation control in Navigation Mode section, executed the extra error-control logic.

After the offline experiment, the participants were allowed to take a 15 min break, without taking off the EEG cap. The decoder was trained offline in the meantime. Offline and online experiments took less than 120 min in total, depending on how much time was left for the online experiment; participants navigated in at least one scenario, a museum or a city, using low-level and middle-level control, respectively (details described in the Online Experiments session). In addition to the two MI tasks, the eye blinking signal, a typical artifact, was also used as an extra control signal in the study, see Navigation control in Navigation Mode section.

### 3.5. BCI Decoder

The decoder in this study is based on a multi-frequency filter bank following a common spatial pattern approach (FBCSP) [30]. As illustrated in Figure 4, data processing consists of three steps. First, the 6-channel raw EEG signal is band-pass filtered into 20 frequency bands using 5th order Butterworth filters, leading to an output with 120 channels. Starting from 4 Hz, each frequency band has a pass band of 4 Hz and overlaps by 2 Hz with the last frequency band until the frequency band that starts at 40 Hz, and finally, there is a 4–40 Hz broadband filter. Secondly, the signals from each band are applied to a CSP filter [31] that maximizes the variance of either the non-control state and the two-hand flexion state, or the left and right-hand clenching, depending on the task. Each CSP filter spatially compresses the frequency band data from six channels to two channels. Thus, the output of the CSP stage has 40 channels.

Next, features are extracted from the CSP-filtered signals. Similarly to [32], the statistical features listed in Table 1 are extracted in a sliding window way. A window with 200 time steps, and a hop size of 15 is applied to each channel, inside which features are calculated. As 5 features are extracted from each of the 40 CSP channels, the final feature map has 200 dimensions, for which feature scores are available for 117 time steps for the F-IM task and 67 for the L/R-IM task. Finally, the resulting feature-time matrix is classified by a bidirectional long short term memory (BiLSTM) network [33], which is based on LSTM [34], a particular type of recurrent neural network (RNN) specialized in solving the gradient descent problem when dealing with long-sequence data. The BiLSTM is believed to capture the underlying context better than the LSTM by traversing the input data forward and backward [35]. The proposed network structure is shown in Figure 5. The network was built using the deep learning toolbox in Matlab, with the following layer settings:



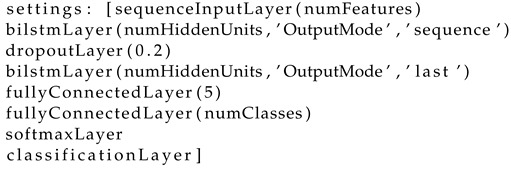



A detailed analysis of the proposed decoder can be found in Figure A1, which was generated by MATLAB’s analyzeNetwork function. During training, an Adam optimizer was used, with an initial learning rate of 0.00005 that dropped every iteration by a factor of 0.8. To control the problem of overfitting to a limited number of training examples, the maximum iterations was set to be 50. In addition, we also applied L2 regularization (0.1) and a dropout layer. Since the focus of this paper is not on the decoder, the choice of hyperparameters and network structure was based on a simple hyperparameter selection obtained from the data during few test runs at the beginning of the study. Additionally, in online experiments, first, an artifact rejection step is applied to the raw EEG signal. If the recorded epoch’s maximum value is over 200 μ V, this epoch is regarded as having an artifact and is further not classified. In addition, in the state where eyeblink acts as a control signal (see Navigation Modes), a simple threshold method is used to detect eyeblinks: when the maximum value recorded from Fz (band pass filtered between 4 and 40 Hz) is larger than 100 μ V.

**Figure 5 sensors-23-01704-f005:**
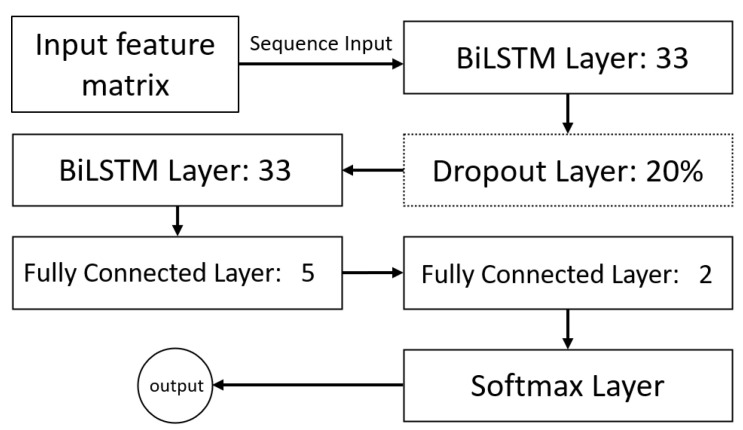
Neural network architecture: the dashed dropout layer is only used in the training phase.

**Table 1 sensors-23-01704-t001:** Overview of the extracted features.

Features	Formula	Remark
The logarithmic sum of power (p)	$p=log(\frac{1}{n}*\sum_{i=0}^{n}x_{i}^{2}\;+1)$	n: the number of samples
The logarithmic of variance (v)	$v=log[\frac{1}{n}*\sum_{i=0}^{n}{\left(x_{i}-\mu \right)}^{2}\;+1]$	n: the number of samples
Zero crossing rate (ZCR)	$ZCR=\frac{1}{2W_{L}}*\sum_{n=1}^{W_{L}}\left|sgn\left[x_{i}\;\left(n\right)\right]-sgn\left[x_{i}\left(n-1\right)\right]\right|$	$W_{L}$: the length of the signal, n: the number of samples, $|sgn\left[x_{i}\;\left(n\right)\right]=\left\{\begin{array}{cc} 1 & x_{i}\left(n\right)\geq 0\\ -1 & x_{i}\left(n\right)<0 \end{array}\right.$
Sum of absolute value (SA)	$SA=\sum_{i=0}^{n}Abs\left(x_{i}\right)$	n: the number of samples
Peak to peak value (p2p)	$p2p=x_{max}-x_{min}$	$x_{max}$: the maximum value of samples, $x_{min}$: the minimum value of samples

### 3.6. Application Interface

The proposed navigation application interface is shown in Figure 6. It consists of two elements: the main window, marked in red in panel A, which shows the Google street view^®^ of the Rijksmuseum, the Netherlands [36], is based on the open-source Moving-AI project [37]; the state bar, marked in yellow in panel B, shows the current state and gives instructions on allowed control commands based on the current state. More specifically, as shown in panel C, four actions can be performed by users: two-hand flexion (Flexion), left-hand clenching (Left), right-hand clenching (Right), and eye blinking (Blink). Depending on the current state, only some actions are mapped to navigation commands, represented by the images below. For example, in state A, one can perform flexion to start walking or switch to the rotation mode by blinking; in state B, one can perform left or right-hand clenching to rotate and stop rotating by eye blinking.

### 3.7. Navigation Mode

The control block diagram of the proposed low- and middle-level navigation modes is shown in Figure 7a. The user starts the application by performing the two-hand flexion task; once detected, the user enters the switch state, a short 2 s transition state that determines whether the user intends to move or rotate. When an eye blink is detected in this state, the system transits to the rotation state, in which the user performs cue-based single-hand clenching to control left or right rotation with different error-control strategies (Figure 7b). Without doing anything, the system enters the walk state, in which case the user moves forward every second. To stop walking, the low-level mode detects the user’s eye blink, and the middle-level control mode is based on the user’s current geographical location: it stops in front of a junction. The middle-level control is accomplished by querying the current position from the GeoNames Web Service [38].

Figure 7b demonstrates two error-control strategies deployed in the rotation state: a double confirmation scheme (DC) and an error correction scheme (EC). Every time the decoder detects a rotation intention for the first time, an arrow is displayed on the screen pointing to the classified direction. If the DC strategy is used, the user must imagine the same hand clenching twice if the direction is the intended one. Only if both attempts are correctly classified, rotation is executed. As for the error correction scheme, an extra correction state is present between the arrow displayed on the screen and the execution of the rotation. The user can blink in this state if the arrow is not pointing to the intended direction. When an eyeblink is detected, the opposite arrow direction will be executed.

### 3.8. Online Experiments

Before the online experiment, the task was detailed to the participant during the inter-session breaks. First, participants learned how to control the interface. Then, a printed map indicating the predefined navigation path was shown and placed in front of them. During the online task, participants were told to follow that path and to navigate in a Google Street View application using various strategies (see below). To evaluate the online experiment, we report whether participants succeeded in reaching the destination, the online control accuracies, and the time to accomplish the task.

First, six experiments were conducted on three participants using no error control strategy (NE) to control the rotation. This experiment was done prior to the other navigation experiments to assess feasibility. Different error-control strategies were proposed, developed, and tested on a larger population given the poor performance obtained using the NE strategy.

To verify that the proposed system could be used for navigation, 4 participants were given the low-level museum task (1 in the Rijksmuseum [36] using DC and 3 in the Uffizi gallery [39] using DC), and 18 were given the middle-level city task, in which different error-control strategies were evaluated. In total, 17 experiments were conducted using DC, and 7 were conducted using EC. Notice that one participant could perform multiple tasks when enough time is available.

To assess the low-level control scheme, the participant was asked to navigate either the Uffizi Gallery in Italy [39] or the Rijksmuseum in the Netherlands [36], following a specific path and stopping at different artworks, as shown in Figure 8. The dark blue line marks the task trajectory that participants needed to follow, starting from the green marker and ending at the red flag. Participants were required to stop at white circle positions and control the rotation to the next task point. As for the middle-level control, the participant was asked to reach the Leuven Town Hall from Diestsestraat, the main shopping street of Leuven, Belgium. More details will be discussed in the results session (Task Completion Overview).

### 3.9. Evaluation Metrics

After offline experiments, the two decoders were trained on the spot and evaluated in terms of testing data accuracy. For online experiments, we report the following results. First, what was displayed on the screen during the online experiment was recorded. After that, the numbers of correct and wrong trials were counted based on the screen recordings, and the decoder’s accuracy was determined. In addition, we evaluated the total time to complete the task, the time needed to issue each navigation command under different control strategies, and the number of mistakes the system made in the middle-level control scheme. Since the time measurement was not normally distributed, the Wilcoxon rank sum test was performed to test if there was any statistical significance between different error control strategies.

## 4. Results

### 4.1. CSP Patterns

Similarly to [40], the spatial patterns for the Mu (8–12 Hz) and the beta bands (18–22 Hz) are plotted in Figure 9 for three randomly selected participants and the pattern averaged across all participants. We report the two patterns that maximize the variance of the imagined left-hand clenching and right-hand clenching, respectively. In general, the averaged pattern has a similar contralateral pattern to the one reported in [40] (Figure 5) that has obvious negativity in the left hemisphere for the right-hand IM task, and in the right hemisphere for the left-hand IM task.

### 4.2. Decoder Performance

Offline performance is shown in Figure 10 in terms of accuracies of the two-hand flexion–resting and left–right-hand-clenching classifiers. Blue lines indicate the maximum, mean, and minimum accuracies obtained from different subjects. On average, roughly 80% and 63% accuracies are achieved overall for the flexion–rest and left–right classes, respectively. Online performance was evaluated differently for the flexion–rest classifier and the left–right classifier. Since two-hand flexion was used as a voluntary start signal, the time needed for the classifier to detect the intention is reported in the next section. As for the left–right classifier, the number of correctly classified trials and the number of wrongly classified trials were counted and are reported in Figure 11. The labels “Left” and “Right” indicate the success rates of correctly identifying the left and right-hand clenching intentions. “Acc” is the average rate for correct detections. No significant difference was found between offline and online accuracies (*p* = 0.91, Wilcoxon signed rank test). However, online accuracies exhibited greater variability across subjects, with the highest accuracy being 1 and the lowest 0.35, whereas the highest and lowest accuracies from offline decoders were 0.854 and 0.42, respectively.

### 4.3. Navigation Performance

#### Task Completion Overview

Statistics regarding the online experiments are listed in Table 2. As described in the earlier session, six experiments of no-error-control rotation control were first done on three participants. Regarding navigation control, one participant performed at least one experiment of low-level control or middle-level control using DC or EC strategies.

In Figure 12, we report the performance and task completion times of all participants in the online navigation task. The dark blue line marks the task trajectory that participants needed to follow. Participants needed to control navigation at each junction, indicated by white circles, starting from the green circle and ending at the red flag. The time taken to pass through each junction is shown in the orange bar; three numbers indicate the minimum, average, and maximum numbers of seconds participants spent at that junction. Light-blue arrows indicate mistakes made by two subjects when navigating the city. The two failed attempts are reported by red crosses in white-filled circles, representing the last positions when the two participants decided to stop the experiment. Some experiment screenshots can be found in Appendix A

### 4.4. Navigation Time Comparison

The lengths of time needed to complete the online experiment, to issue a forward walking command and an intended rotation command with different error-control strategies, are reported separately in Figure 13 and Figure 14. The Wilcoxon rank sum test results are listed in Appendix A Table A1, where red numbers indicate *p*-values smaller than 0.05.

### 4.5. Rotation Accuracy

As to correctly issuing one online rotation command, we also report how many trials of imagination the system misclassified (No_error), how many wrong rotations the system performed (No_navigation_error), and how many attempts the user needed to make before the intended rotation was executed (No_imagination). The difference between No_error and No_navigation_error is that the former did not necessarily result in an unwanted rotation, whereas the latter always refers to an unwanted rotation. The results are listed in Table 3. The Wilcoxon rank sum test results are listed in Appendix A Table A2–Table A4, where red numbers indicate *p*-values smaller than 0.05.

### 4.6. User Questions during Online Navigation

User feedback is important to identify needs, preferences, and frustrations, and to identify potential improvements. Albeit we did not formally query our users about the system’s usability, such as being intuitive, easy to use, or effective, we took note of which and how many questions they asked during the online experiment. Before the online experiment, each participant was instructed orally about the control of the system in the online experiments during the two breaks of the offline training sessions, for about 10 min in total. In the online experiments, 7 out of 21 subjects finished the task without asking any questions regarding the control, but for the other 14 participants, an average of 4.43 questions were asked during the 20 min session. The maximum of eight questions came from one subject who failed to complete the task. Participants asked questions because they were struggling with the system and needed guidance, or when the system made mistakes of which they thought they were the culprits, which made them doubt they were in control.

## 5. Discussion

According to [41], a high information transfer rate is a primary requirement of an effective BCI application but is hampered by the inherently low SNRs of EEG signals. Although clinical-grade gel-based electrodes, densely distributed over the scalp, can yield superior performances, with their long preparation time, use cases are limited to laboratory settings. While with wireless EEG recording devices often have a restricted number of dry-electrodes, preparation is fast and easy, and suitable for daily use. We believe that developing BCI applications for dry-electrode devices can move BCIs out of the laboratory into the real world. In same vein, that study developed a MI-BCI navigation application that addresses poor decoder performance due to the use of a limited number of dry electrodes by system design and the use of error-control strategies. The resulting CSP patterns imply that the traditional CSP method may perform worse when using only a few electrodes, as it may fail to capture the spatial features that could vary between subjects. As we targeted an EEG set-up with a restricted number of dry-electrodes and refrained from using a more sophisticated feature set, we stressed the importance of the whole BCI system’s design. We proposed a new middle-level control scheme to fill the inefficiency gap in VE navigation. Overall, the over 80% success rate of online experiments was achieved, except for the NE scheme, confirming the feasibility of an MI-controlled BCI navigation system. In addition, the middle-level navigation scheme demonstrated an efficient way of navigating in an unknown virtual environment. According to Figure 12, users only needed to intervene at road junctions. Although the middle-level control was tailored to VE navigation, it sheds light on novel real-world navigation methods. Thanks to the advances in computer vision and autonomous driving systems, junction detection could be implemented in real life, thereby making the proposed middle-level control applicable to real-world navigation.

The proposed system differs from existing VE navigation systems in two ways. First, the proposed Google Street View^®^ application offers a higher degree of immersion to users. Typical BCI-VE navigation features a specific scene only, e.g., a virtual library [25], a virtual street [12], or a virtual apartment [19]. The scenes in each system are monotonous, either indoor or outdoor. On the contrary, in Google Street View^®^, users can choose themselves to visit a city, museum, shopping mall, zoo, etc. Second, the proposed control logic has wider applicability. In many studies, the control logic does not easily transfer to other navigation tasks. For example, the system in [12] can only be used for rotation or walking forward, not both in combination. The system in [19] only works with two options when at a junction. Although the low-level control logic can rely on visual paradigms only, as in [6,7,14,15,16,17,18], it is rather impractical to navigate a city with.

According to Figure 10 and Figure 11, the obtained accuracies of the developed decoders were rather low, especially compared with the reported 99% accuracy when using a P300 visual paradigm [6], or the over 91% when using an SSVEP paradigm [7]. Our results confirm the difficulty of decoding MI tasks. Unlike the sophisticated training techniques used in [42] or the two more distinguishable MI tasks (foot vs. hand) in [43] to boost the system’s accuracy, our focus is on system design. The flexion–rest decoder yielded a higher accuracy compared to the left–right decoder, which could be explained by the fact that the task’s duration was much longer (6 s vs. 2 s) and the features were more discriminable (SMR–nothing vs. left SMR–right SMR). The choice was made in view of the trade-off between accuracy and time. As the flexion–rest decoder is used for voluntary start and walking and the left–right is used for controlling rotation, it is less critical to have a longer start time than a longer rotation time, which is performed more often and usually in a repetitive manner.

According to Figure 10 and Figure 11, the large variability in accuracies found in online experiments shows the intra-subject decoder variability, and the increase in accuracy implies that some participants could adapt to the decoder. The large subject-dependent variability can be also observed in Figure 9; each subject had different CSP patterns corresponding to the imagined left and right-hand clenching. In general, the averaged patterns showed a contralateral behavior that had negativities in CZ, C3, and P3 for the right-hand IM tasks; and in C4 and P4 for the left-hand IM tasks. However, the three subjects exhibited different patterns, some of which were similar to the averaged ones, and others even opposing (cf. the right pattern for subject b). Apart from the addressed subject variability issue, this could be due to the limited number of recording channels. Indeed, as can be observed in Figure 5 in [40], some patterns were tightly clustered at a few channels in the frontal and temporal regions, which were not covered in our recording setup. The limitation in available electrodes could explain why some subject-specific patterns were not captured, which in turn could explain the large variability in accuracy among subjects. This reflects the challenge of decoding IM tasks using limited dry electrodes, which together with the low SNR of dry electrodes, calls for decoding algorithms that can deal which such setups.

The results also indicate that BCI control could still work, even with low decoder accuracy, through a well-designed control scheme. With an error-control strategy, the number of unwanted rotations was significantly reduced, as can be seen in Appendix A Table A3. However, as a compromise, more time was needed to issue an intended navigation command, as can be observed in Figure 12 and Figure 13. Ideally, 100% accuracy system would have a command issue time equal to a single epoch length: 6 s to start walking and 2 s to rotate. However, actual walking time was 3 times larger, and rotation time was at least 10 times larger, than the ideal cases, when the decoder had a true positive rate of 100%.

Comparing different error-control strategies, according to Figure 13, EC was significantly faster than DC, which we believe was caused by (1) the number of errors being significantly smaller (Appendix A Table A2) and (2) the number of imaginings being significantly smaller (Appendix A Table A4). The latter was due to the nature of EC: as long as the eye blinking detection was accurate, the number of imaginings needed was only one, and the error correction mechanism could correct the mistake made by the decoder. In contrast, at least two imaginings were needed for DC, and if a mistake was made by the system, two more imaginings were needed to correct it. This can be proven in Figure 14, where the rotation time for DC is significantly larger than the EC. Meanwhile, with fewer imaginings subjects have to perform, fewer errors are made by the system, demoting the occurrence of errors. Interestingly, DC had no significant difference from SC in any metrics, except for the number of navigation errors. The interpretation is straightforward: without any improvement in the decoder itself, the DC could only avoid unwanted rotations, but in the meantime, it would also take more time for two consecutive trials to be correctly classified. As a result, the number of imaginings the decoder needed, the number of mistakes it made, and the time it took, were comparable to those of the NE strategy. Furthermore, in Table 3, several facts can be observed. First, for the NE strategy, the number of classification errors equaled the number of navigation errors, since each output from the decoder resulted in a navigation movement. To correct the errors the system made before, plus to rotate towards the desired direction, the number of total imaginings was more than twice the number of mistakes. As for the DC strategy, the number of imaginings needed was even larger, since forcing the same two trials consecutively not only lowered the chance of making wrong turns but also made it more difficult to issue a wanted rotation. On the other hand, the EC strategy showed the best performance. With a significantly small number of imaginings, one can rotate in the desired direction using the EC strategy. Overall, the results implied the importance of BCI system design and the necessity of error-control strategies.

In terms of system usability, only one third of participants could learn and memorize the controls through the oral instructions given before the online experiments. For the rest, guidance was necessary, especially when the system made mistakes, which caused many participants to doubt if they had made any mistakes. In some follow-up interviews, when we asked whether the participants would like to use it daily, all answered “no”. The reasons were twofold. On one hand, as we recruited healthy participants only, the proposed BCI system did not offer more benefits than the traditional keyboard-based system, and on the other hand, the amount of training time and the level of concentration during the experiment quickly made them feel tired. Therefore, a more immersive and interesting application may attract healthy users, and a better decoder is crucial to improve user experience. In the future, to better address this problem, a more detailed and quantitative usability test needs to be conducted.

In terms of technical aspects, the proposed system has some limitations. First, Google Street View^®^ only allows the user to navigate along a predefined path; consequently, his level of freedom is constrained. Second, the minimum junction-passing time of 1 s indicated in Figure 12 was caused by the flaw in operation when relying on two online services. The system uses the location provided by Google Maps to query the junction information from GeoNames Web Service [38], a geographical database different from the former. Inconsistencies between the two systems sometimes led to inaccurate junction detection results, as the geographical location that Google treated as a junction might not correspond to a junction in the GeoNames’ database, and vice versa, sometimes causing the system not to stop at a junction. Technically, this could be solved by using a commercial-level geographical database server or the same Google database. In practice, the former means a paid service, whereas the latter is difficult to achieve, as Google does not provide a direct junction query service. However, this middle-level control design pattern can be adapted in virtual applications and games where locations and environments are known to the system. With the metaverse concept being in the spotlight, a post-reality universe provides users with real-time interaction with a virtual environment and with other users [44]. We believe this study can shed light on integrating BCI-based control into VR applications—for example, for controlling a virtual avatar in VRChat [45], a VR-based community that enables users to create their own virtual worlds, either an indoor environment or in a city-like open environment, just like Google Street View^®^.

Although we proposed a working BCI navigation application with two error-control strategies, to compensate for the poor performance of the MI decoder, some interesting research topics are worth further exploring. Firstly, this study only tested the method on 21 healthy subjects for feasibility verification. To fully understand the usability of the proposed system in the future, more healthy subjects will need to participate, along with paralyzed individuals. Usability tests will be conducted to identify potential issues with the system. Secondly, the poor decoding accuracies obtained by using CSP call for the future development of new MI decoding algorithms that suit the dry-electrode setups. Thirdly, high-accuracy multi-class MI decoders could lead to better performing navigation designs, and faster navigation could be achieved with more straightforward control logic; for example, a 5-class MI based navigation system was used to control a wheelchair in [9]. Unfortunately, the authors did not report either the accuracy of their decoders or their online evaluation results. Furthermore, when targeting mobile EEG devices, removing movement artifacts can lead to better EEG signal quality and improved performance [46]. Another possible improvement could be to tailor the proposed decoder to individual subjects by using some form of automatic hyperparameter tuning—for example, applying Bayesian optimization for feature selection [47] and hyperparameter optimization for the network structures [48]. Moreover, other BCI paradigms might lead to more user-friendly designs; for example, one could use imagined speech to switch between walking, rotation, and standing, while using IM for rotating only. Thirdly, the results of the comparison between different error-control strategies indicated the importance of detecting errors with BCI systems. The proposed error-detection mechanism relies on eye blinks, an active interaction between users and machines, whereas a passive error detection system, as used in some communication BCI systems (e.g., [49,50]), is believed to be able to increase the information transfer rate, based on error-related potentials [51], which has not yet been introduced in a navigation system. A passive error detection BCI might be worth implementing to “close the loop” of the navigation system, further simplifying its design. Finally, a VR-based navigation application can provide a more immersive experience to users.

## 6. Conclusions

This paper reported on the design and implementation of an MI-VE application to Google Street View^®^, a further step towards real-life BCIs. The system was tested on 21 healthy participants with both offline and online experiments to navigate in museums and a city. The experiments verified the feasibility of MI-VE navigation using a wireless, dry electrodes EEG cap, even with poor decoder accuracies of around 80% and 60% on average for flexion–rest and left–right decoders among participants, respectively. Furthermore, a novel middle-level control scheme was proposed, which was designed to offer a balanced trade-off between user control freedom and navigation efficiency. When upgrading to real world junction detection and obstacle avoidance, our design could potentially be used in real-life navigation as well. Finally, experiments were carried out to evaluate different error-control strategies. The results suggest that mistakes could be largely avoided by employing error-control strategies, calling for a compromise with control duration. The proposed error correction through eye blinks returned the best performance, suppressing the number of navigation mistakes with an acceptable increase in control time. In conclusion, this paper demonstrates the importance of system design with respect to BCI-based control and calls for more effort on developing VE-BCI navigation applications and better system control designs.

## Figures and Tables

**Figure 1 sensors-23-01704-f001:**
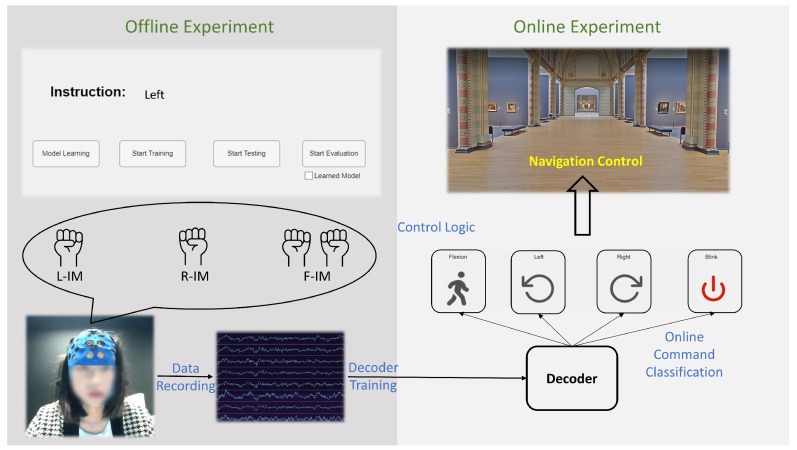
System Overview: **left** panel: offline training session, **right** panel: online real time navigation control. L-IM stands for the left-hand imagined motor task, R-IM for the right-hand imagined motor task, F-IM for the flexion imagined motor task. See text for explanation. (The participant has given the consent to use her blurred image).

**Figure 2 sensors-23-01704-f002:**
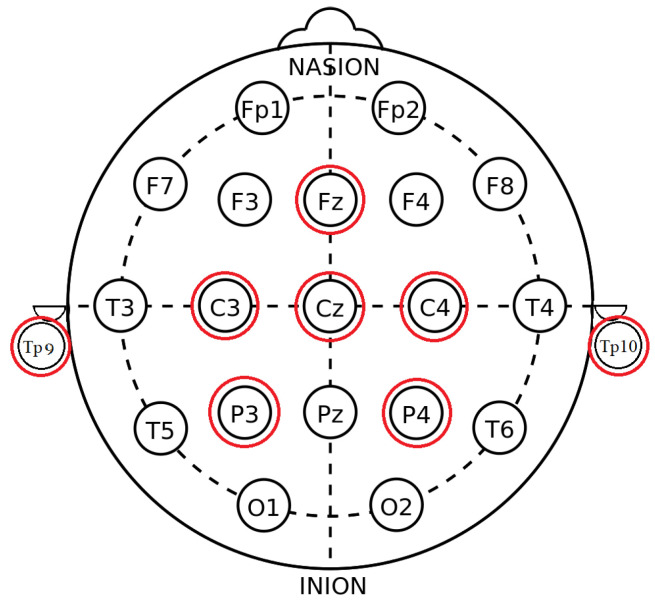
Electrodes’ locations.

**Figure 3 sensors-23-01704-f003:**
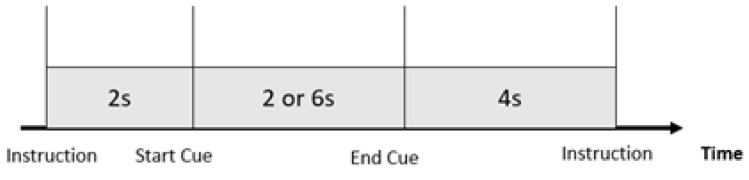
Timing of the training session.

**Figure 4 sensors-23-01704-f004:**
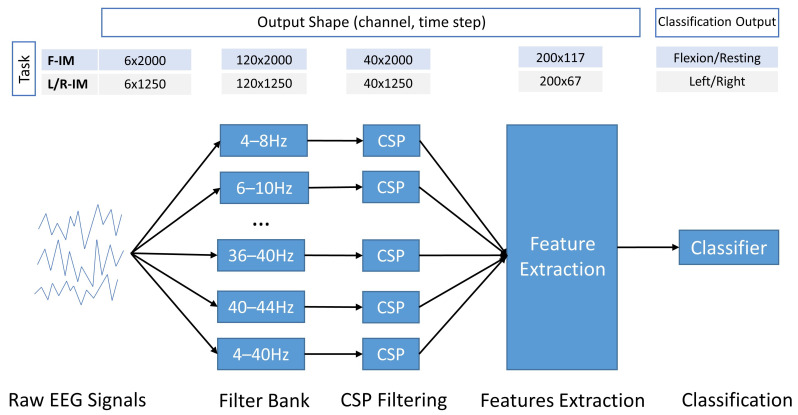
EEG processing pipeline: the output shape after each step is listed in the above table.

**Figure 6 sensors-23-01704-f006:**
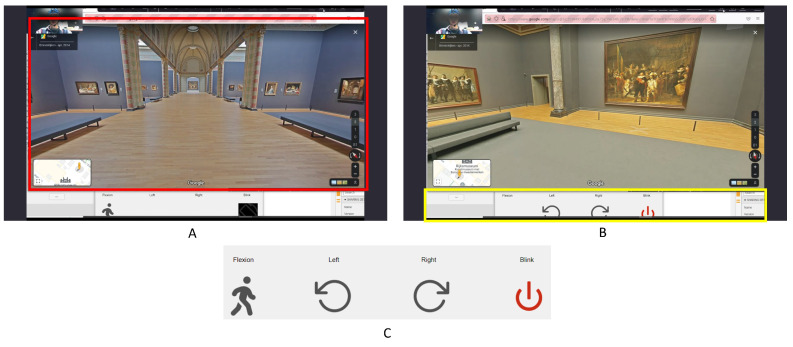
Navigation interface. Panel A: the main navigation window, Panel B: the state bar, Panel C: the four navigation actions.

**Figure 7 sensors-23-01704-f007:**
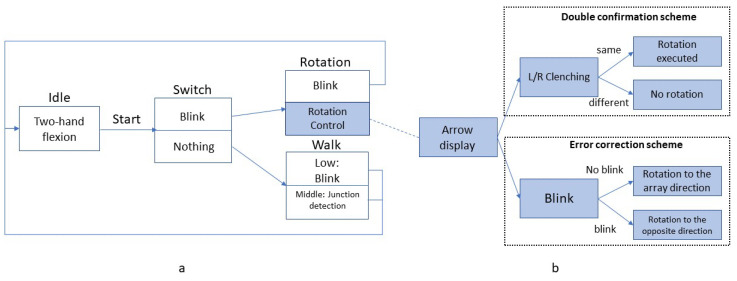
Navigation Control Diagram. (**a**): low- and middle-level control modes, (**b**): error control strategies.

**Figure 8 sensors-23-01704-f008:**
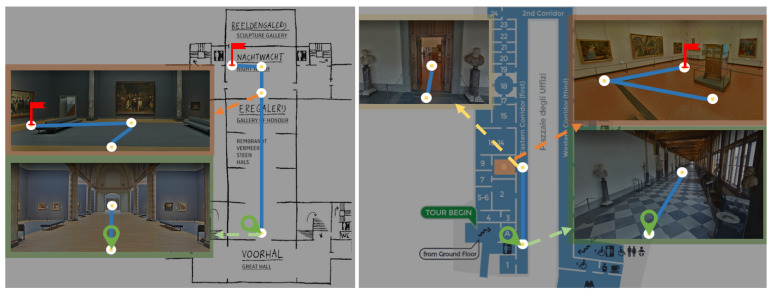
Low level navigation task, the Rijksmuseum on the **left**, the Uffizi Museum on the **right**. The layout maps are from https://www.visituffizi.org/museum/uffizi-floor-plans/ and https://kalden.home.xs4all.nl/mann/Mannheimer-inrijksmuseum.html (accessed on 20 November 2022) See text for explaination.

**Figure 9 sensors-23-01704-f009:**
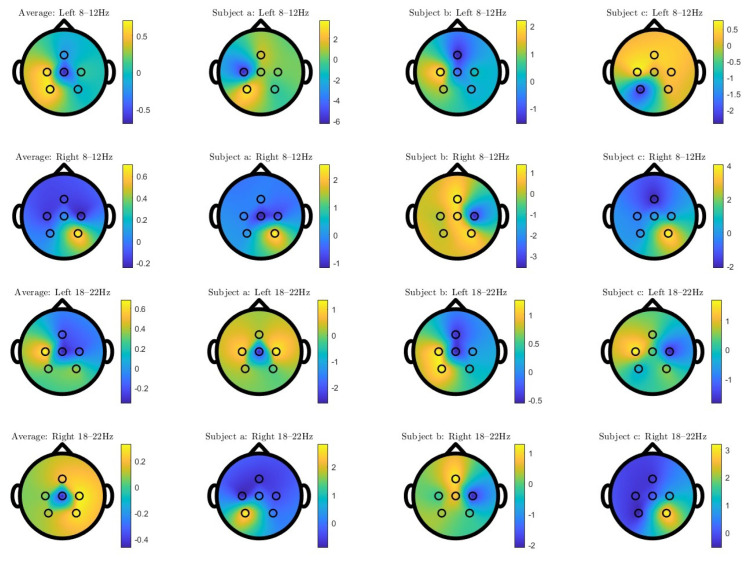
Spatial patterns corresponding to the 8–12 and 18–22 Hz bands (top 2 rows and bottom 2 rows), averaged for all participants (Average) and for 3 randomly selected subjects (subjects a, b, and c) (arranged columnwise), for imagined left-hand clenching (**Left**) and imagined right-hand clenching (**Right**) (arranged top and bottom row in each row-pair).

**Figure 10 sensors-23-01704-f010:**
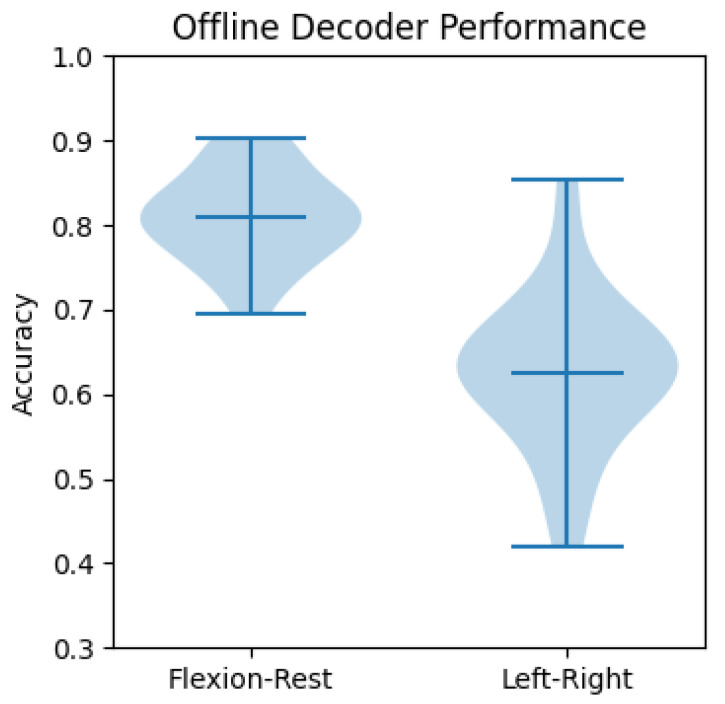
Offline decoder’s performance.

**Figure 11 sensors-23-01704-f011:**
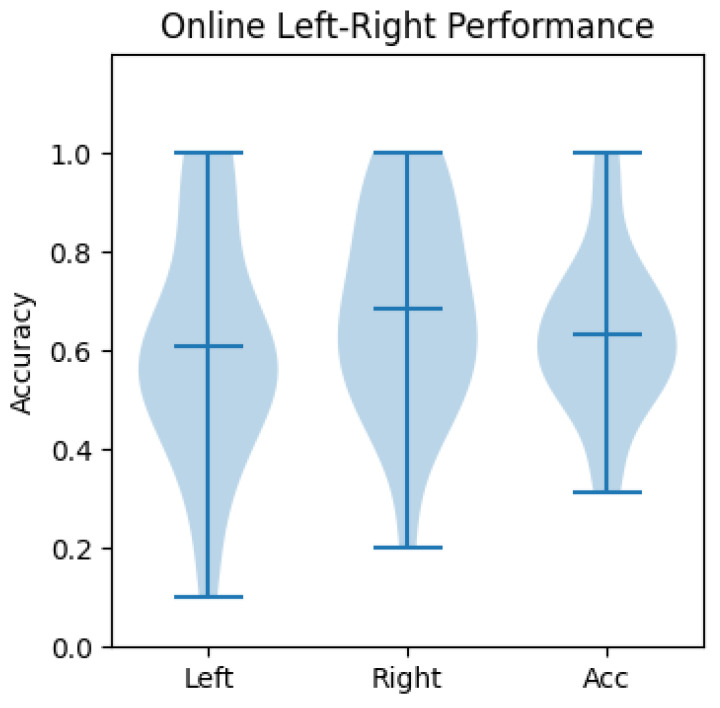
Online decoder’s performance.

**Figure 12 sensors-23-01704-f012:**
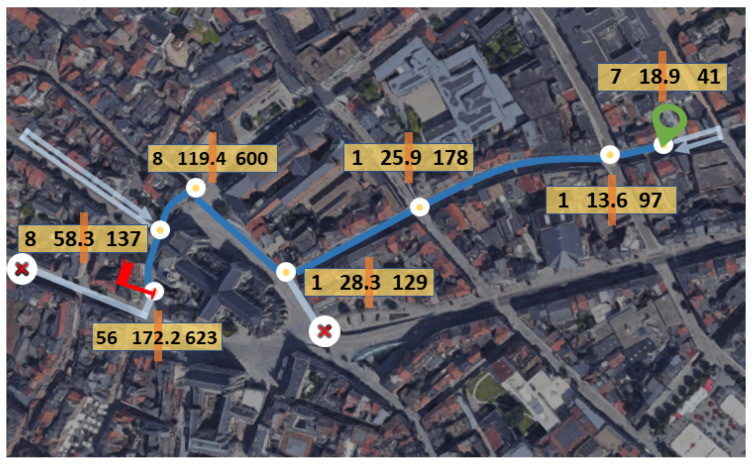
Task completion overview. See text for explanation.

**Figure 13 sensors-23-01704-f013:**
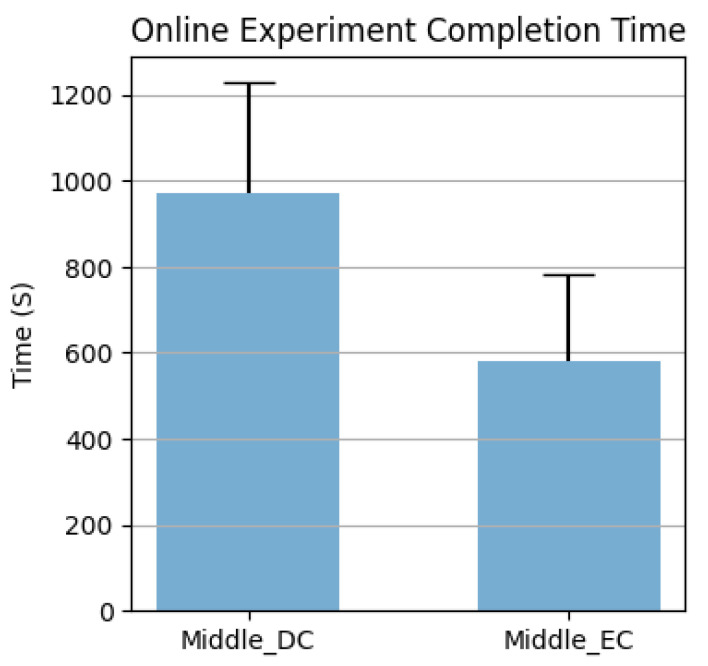
Time to complete the middle-level experiment with DC and EC strategies.

**Figure 14 sensors-23-01704-f014:**
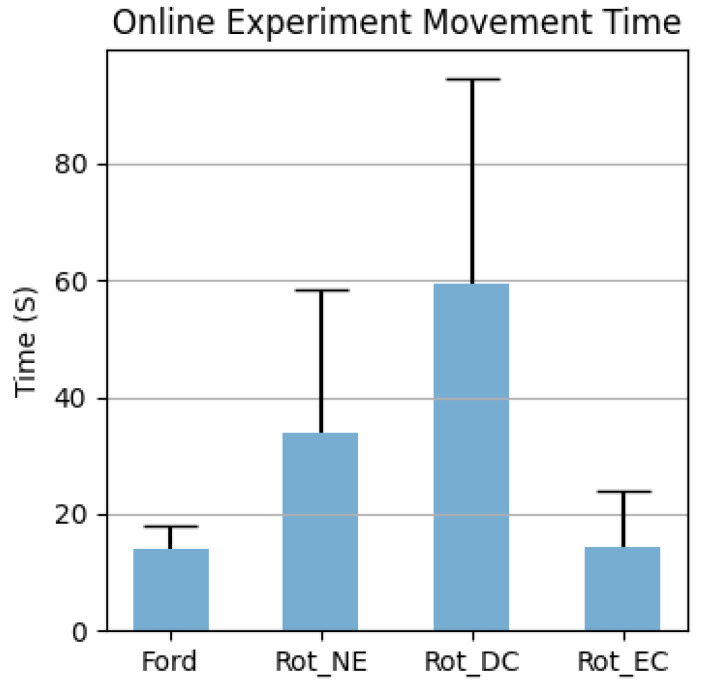
Time to issue a correct navigation command. Ford stands for moving forward, Rot for rotation, dc for double confirmation strategy, EC for error correction strategy, and NE for no error control strategy.

**Table 2 sensors-23-01704-t002:** Overview of online experiments.

Count	Low-Level Control	No Error Control	Middle-Level-DC	Middle-Level-EC
Total number of experiments	5	6	17	7
Success trails	4	3	15	6
Success rate	0.80	0.50	0.882	0.857

**Table 3 sensors-23-01704-t003:** Overview of online rotation accuracy using different error-control strategies.

Error Control Strategy	No_Error	No_Navigation_Error	No_Imagination
Rot_S	1.34	1.34	4.48
Rot_DC	1.87	0.29	6.79
Rot_EC	0.70	0.32	1.63

## Data Availability

The data are not publicly available but can be made available upon request.

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
