# Peer review of "Real-Time Navigation in Google Street View® Using a Motor Imagery-Based BCI"

_sensors, 2023, doi:10.3390/s23031704_

Round 1

Reviewer 1 Report

• What is the main question addressed by the research?

• Do you consider the topic original or relevant in the field? Does it
address a specific gap in the field?

 The authors proposed a novel MI-BCI application based on an 8 dry-electrode EEG setup, with which users can explore and navigate in Google Street View. Yes, the authors have addressed the poor performance of the MI decoder due to the dry electrodes’ lower signal quality. Also, the authors made restrictions on the number of navigation commands by using a novel middle-level control scheme. The authors have used eye blinks as a control signal in different navigation stages.

• What does it add to the subject area compared with other published
material? 

Typically, existing methods have focussed on low and high level control strategies whereas the authors have introduced middle level control scheme.

• Are the conclusions consistent with the evidence and arguments presented
and do they address the main question posed?  

yes

• Are the references appropriate?  

yes

• Please include any additional comments on the tables and figures.  

looks ok.  

The authors are suggested to include the permission obtained

from ethical committee in the declaration section.

Author Response

We thank the reviewer for his/her comments, which we address in the word file

Reviewer 2 Report

The research manuscript presents a study on the use of Brain-Computer Interfaces (BCIs) for navigation in virtual worlds. The authors propose a novel Motor Imagery-based BCI (MI-BCI) application that allows users to explore and navigate in Google Street View® using a 8 dry-electrode EEG setup. The study aims to address the limitations of previous BCI-based navigation studies which adopted cue-based visual paradigms and to explore the use of MI-BCI in future games and virtual reality (VR) applications for consumers as well as for patients temporarily or permanently devoid of muscle control.

The study presents a number of contributions to the field of BCI-based navigation. Firstly, the authors propose a novel middle-level control scheme that restricts the number of navigation commands, addressing the lower performance of the MI decoder due to the dry electrodes’ lower signal quality and the low number of electrodes. Secondly, the authors introduce eye blinks as a control signal in different navigation stages in order to avoid decoder mistakes. Thirdly, the study is conducted with 20 healthy subjects and the results show acceptable performance, even given the limitations of the EEG set-up, which the authors attribute to the design of the BCI application. The composition of subjects shows gender imbalance, please explain if it had any impact on the results.

However, the study also has several limitations. The use of dry electrodes, which have lower signal quality compared to wet electrodes, is one of the main limitations. Additionally, the sample size of 20 healthy subjects is relatively small and the study would benefit from a larger sample size. Moreover, the study only used Google Street View® for navigation, which limits the usability of the proposed system for other types of virtual worlds.

The BCI system  is intended to be used during physical navigation, which raised the need to consider and use appropriate filters to remove movement artefacts. The authors are encouraged to consult doi:10.1109/ACCESS.2018.2890335 for further details.

The performance of the system is evaluated using accuracy metric. However, the usability aspect of the system is missing. Discuss the usability of the developed system, and present its evaluation using SUS or other similar method, if possible.

In conclusion, the study presents a novel MI-BCI application for navigation in virtual worlds that addresses some of the limitations of previous BCI-based navigation studies. However, the use of dry electrodes and a small sample size are limitations that should be considered when interpreting the results of the study. 

Author Response

We thank the reviewer for his/her comments, which we address below in the word file

Round 2

Reviewer 2 Report

The authors have revised well. The manuscript can be accepted for publication.

Author Response

We thank the reviewer for his valuable suggestions and we would like to notify that more changes have been made. We refer to the cover letter for details.